# Structured Embedding Models for Grouped Data

**Maja Rudolph**
Columbia Univ.
maja@cs.columbia.edu

**Francisco Ruiz**
Univ. of Cambridge
Columbia Univ.

**Susan Athey**
Stanford Univ.

**David Blei**
Columbia Univ.

## Abstract

Word embeddings are a powerful approach for analyzing language, and exponential family embeddings (EFE) extend them to other types of data. Here we develop structured exponential family embeddings (S-EFE), a method for discovering embeddings that vary across related groups of data. We study how the word usage of U.S. Congressional speeches varies across states and party affiliation, how words are used differently across sections of the ArXiv, and how the co-purchase patterns of groceries can vary across seasons. Key to the success of our method is that the groups share statistical information. We develop two sharing strategies: hierarchical modeling and amortization. We demonstrate the benefits of this approach in empirical studies of speeches, abstracts, and shopping baskets. We show how S-EFE enables group-specific interpretation of word usage, and outperforms EFE in predicting held-out data.

## 1 Introduction

Word embeddings (Bengio et al., 2003; Mikolov et al., 2013d,c,a; Pennington et al., 2014; Levy and Goldberg, 2014; Arora et al., 2015) are unsupervised learning methods for capturing latent semantic structure in language. Word embedding methods analyze text data to learn distributed representations of the vocabulary that capture its co-occurrence statistics. These representations are useful for reasoning about word usage and meaning (Harris, 1954; Rumelhart et al., 1986). Word embeddings have also been extended to data beyond text (Barkan and Koenigstein, 2016; Rudolph et al., 2016), such as items in a grocery store or neurons in the brain. Exponential family embeddings (EFE) is a probabilistic perspective on embeddings that encompasses many existing methods and opens the door to bringing expressive probabilistic modeling (Bishop, 2006; Murphy, 2012) to the problem of learning distributed representations.

We develop structured exponential family embeddings (S-EFE), an extension of EFE for studying how embeddings can vary across groups of related data. We will study several examples: in U.S. Congressional speeches, word usage can vary across states or party affiliations; in scientific literature, the usage patterns of technical terms can vary across fields; in supermarket shopping data, co-purchase patterns of items can vary across seasons of the year. We will see that S-EFE discovers a per-group embedding representation of objects. While the naïve approach of fitting an individual embedding model for each group would typically suffer from lack of data—especially in groups for which fewer observations are available—we develop two methods that can share information across groups.

Figure 1a illustrates the kind of variation that we can capture. We fit an S-EFE to ArXiv abstracts grouped into different sections, such as computer science (**cs**), quantitative finance (**q-fin**), and nonlinear sciences (**nlin**). S-EFE results in a per-section embedding of each term in the vocabulary. Using the fitted embeddings, we illustrate similar words to the word INTELLIGENCE. We can see that how INTELLIGENCE is used varies by field: in computer science the most similar words include ARTIFICIAL and AI; in finance, similar words include ABILITIES and CONSCIOUSNESS.

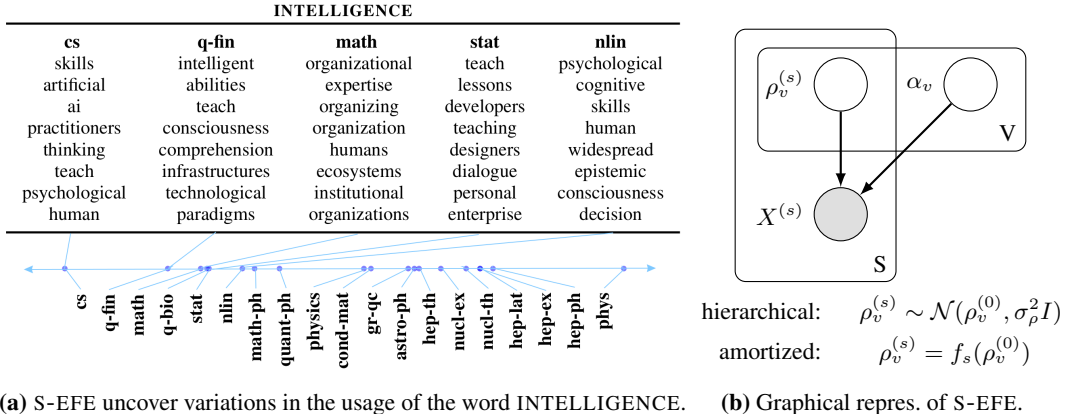

**(a)** S-EFE uncover variations in the usage of the word INTELLIGENCE.     **(b)** Graphical repres. of S-EFE.

**Figure 1: (a)** INTELLIGENCE is used differently across the ArXiv sections. Words with the closest embedding to the query are listed for 5 sections. (The embeddings were obtained by fitting an amortized S-EFE.) The method automatically orders the sections along the horizontal axis by their similarity in the usage of INTELLIGENCE. See Section 3 additional for details. **(b)** Graphical representation of S-EFE for data in $S$ categories. The embedding vectors $\rho_v^{(s)}$ are specific to each group, and the context vectors $\alpha_v$ are shared across all categories.

In more detail, embedding methods posit two representation vectors for each term in the vocabulary; an embedding vector and a context vector. (We use the language of text for concreteness; as we mentioned, EFE extend to other types of data.) The idea is that the conditional probability of each observed word depends on the interaction between the embedding vector and the context vectors of the surrounding words. In S-EFE, we posit a separate set of embedding vectors for each group but a shared set of context vectors; this ensures that the embedding vectors are in the same space.

We propose two methods to share statistical strength among the embedding vectors. The first approach is based on hierarchical modeling (Gelman et al., 2003), which assumes that the group-specific embedding representations are tied through a global embedding. The second approach is based on amortization (Dayan et al., 1995; Gershman and Goodman, 2014), which considers that the individual embeddings are the output of a deterministic function of a global embedding representation. We use stochastic optimization to fit large data sets.

Our work relates closely to two threads of research in the embedding literature. One is embedding methods that study how language evolves over time (Kim et al., 2014; Kulkarni et al., 2015; Hamilton et al., 2016; Rudolph and Blei, 2017; Bamler and Mandt, 2017; Yao et al., 2017). Time can be thought of as a type of "group", though with evolutionary structure that we do not consider. The second thread is multilingual embeddings (Klementiev et al., 2012; Mikolov et al., 2013b; Ammar et al., 2016; Zou et al., 2013); our approach is different in that most words appear in all groups and we are interested in the variations of the embeddings across those groups.

Our contributions are thus as follows. We introduce the S-EFE model, extending EFE to grouped data. We present two techniques to share statistical strength among the embedding vectors, one based on hierarchical modeling and one based on amortization. We carry out a thorough experimental study on two text databases, ArXiv papers by section and U.S. Congressional speeches by home state and political party. Using Poisson embeddings, we study market basket data from a large grocery store, grouped by season. On all three data sets, S-EFE outperforms EFE in terms of held-out log-likelihood. Qualitatively, we demonstrate how S-EFE discovers which words are used most differently across U.S. states and political parties, and show how word usage changes in different ArXiv disciplines.

## 2 Model Description

In this section, we develop structured exponential family embeddings (S-EFE), a model that builds on exponential family embeddings (EFE) (Rudolph et al., 2016) to capture semantic variations across groups of data. In embedding models, we represent each object (e.g., a word in text, or an item in shopping data) using two sets of vectors, an embedding vector and a context vector. In this paper, we

are interested in how the embeddings vary across groups of data, and for each object we want to learn a separate embedding vector for each group. Having a separate embedding for each group allows us to study how the usage of a word like INTELLIGENCE varies across categories of the ArXiv, or which words are used most differently by U.S. Senators depending on which state they are from and whether they are Democrats or Republicans.

The S-EFE model extends EFE to grouped data, by having the embedding vectors be specific for each group, while sharing the context vectors across all groups. We review the EFE model in Section 2.1. We then formalize the idea of sharing the context vectors in Section 2.2, where we present two approaches to build a hierarchical structure over the group-specific embeddings.

## 2.1 Background: Exponential Family Embeddings

In exponential family embeddings, we have a collection of objects, and our goal is to learn a vector representation of these objects based on their co-occurrence patterns.

Let us consider a dataset represented as a (typically sparse) matrix $X$, where columns are datapoints and rows are objects. For example, in text, each column corresponds to a location in the text, and each entry $x_{vi}$ is a binary variable that indicates whether word $v$ appears at location $i$.

In EFE, we represent each object $v$ with two sets of vectors, embeddings vectors $\rho_v[i]$ and context vectors $\alpha_v[i]$, and we posit a probability distribution of data entries $x_{vi}$ in which these vectors interact. The definition of the EFE model requires three ingredients: a *context*, a *conditional exponential family*, and a *parameter sharing structure*. We next describe these three components.

Exponential family embeddings learn the vector representation of objects based on the conditional probability of each observation, conditioned on the observations in its *context*. The context $c_{vi}$ gives the indices of the observations that appear in the conditional probability distribution of $x_{vi}$. The definition of the context varies across applications. In text, it corresponds to the set of words in a fixed-size window centered at location $i$.

Given the context $c_{vi}$ and the corresponding observations $x_{c_{vi}}$ indexed by $c_{vi}$, the distribution for $x_{vi}$ is in the *exponential family*,

$$x_{vi} \mid x_{c_{vi}} \sim \text{ExpFam}\left(t(x_{vi}), \eta_v(x_{c_{vi}})\right), \tag{1}$$

with sufficient statistics $t(x_{vi})$ and natural parameter $\eta_v(x_{c_{vi}})$. The parameter vectors interact in the conditional probability distributions of each observation $x_{vi}$ as follows. The embedding vectors $\rho_v[i]$ and the context vectors $\alpha_v[i]$ are combined to form the natural parameter,

$$\eta_v(x_{c_{vi}}) = g\left(\rho_v[i]^\top \sum_{(v',i') \in c_{vi}} \alpha_{v'}[i'] x_{v'i'}\right), \tag{2}$$

where $g(\cdot)$ is the link function. Exponential family embeddings can be understood as a bank of generalized linear models (GLMs). The context vectors are combined to give the covariates, and the "regression coefficients" are the embedding vectors. In Eq. 2, the link function $g(\cdot)$ plays the same role as in GLMs and is a modeling choice. We use the identity link function.

The third ingredient of the EFE model is the *parameter sharing structure*, which indicates how the embedding vectors are shared across observations. In the standard EFE model, we use $\rho_v[i] \equiv \rho_v$ and $\alpha_v[i] \equiv \alpha_v$ for all columns of $X$. That is, each unique object $v$ has a shared representation across all instances.

**The objective function.** In EFE, we maximize the objective function, which is given by the sum of the log-conditional likelihoods in Eq. 1. In addition, we add an $\ell_2$-regularization term (we use the notation of the log Gaussian pdf) over the embedding and context vectors, yielding

$$\mathcal{L} = \log p(\alpha) + \log p(\rho) + \sum_{v,i} \log p\left(x_{vi} \mid x_{c_{vi}}; \alpha, \rho\right), \tag{3}$$

Note that maximizing the regularized conditional likelihood is not equivalent to *maximum a posteriori*. Rather, it is similar to maximization of the pseudo-likelihood in conditionally specified models (Arnold et al., 2001; Rudolph et al., 2016).

## 2.2 Structured Exponential Family Embeddings

Here, we describe the S-EFE model for grouped data. In text, some examples of grouped data are Congressional speeches grouped into political parties or scientific documents grouped by discipline. Our goal is to learn group-specific embeddings from data partitioned into $S$ groups, i.e., each instance $i$ is associated with a group $s_i \in \{1, \ldots, S\}$. The S-EFE model extends EFE to learn a separate set of embedding vectors for each group.

To build the S-EFE model, we impose a particular parameter sharing structure over the set of embedding and context vectors. We posit a structured model in which the context vectors are shared across groups, i.e., $\alpha_v[i] \equiv \alpha_v$ (as in the standard EFE model), but the embedding vectors are only shared *at the group level*, i.e., for an observation $i$ belonging to group $s_i$, $\rho_v[i] \equiv \rho_v^{(s_i)}$. Here, $\rho_v^{(s)}$ denotes the embedding vector corresponding to group $s$. We show a graphical representation of the S-EFE in Figure 1b.

Sharing the context vectors $\alpha_v$ has two advantages. First, the shared structure reduces the number of parameters, while the resulting S-EFE model is still flexible to capture how differently words are used across different groups, as $\rho_v^{(s)}$ is allowed to vary.[1] Second, it has the important effect of uniting all embedding parameters in the same space, as the group-specific vectors $\rho_v^{(s)}$ need to agree with the components of $\alpha_v$. While one could learn a separate embedding model for each group, as has been done for text grouped into time slices (Kim et al., 2014; Kulkarni et al., 2015; Hamilton et al., 2016), this approach would require ad-hoc postprocessing steps to align the embeddings.[2]

When there are $S$ groups, the S-EFE model has $S$ times as many embedding vectors than the standard embedding model. This may complicate inferences about the group-specific vectors, especially for groups with less data. Additionally, an object $v$ may appear with very low frequency in a particular group. Thus, the naïve approach for building the S-EFE model without additional structure may be detrimental for the quality of the embeddings, especially for small-sized groups. To address this problem, we propose two different methods to tie the individual $\rho_v^{(s)}$ together, sharing statistical strength among them. The first approach consists in a hierarchical embedding structure. The second approach is based on amortization. In both methods, we introduce a set of *global* embedding vectors $\rho_v^{(0)}$, and impose a particular structure to generate $\rho_v^{(s)}$ from $\rho_v^{(0)}$.

**Hierarchical embedding structure.** Here, we impose a hierarchical structure that allows sharing statistical strength among the per-group variables. For that, we assume that each $\rho_v^{(s)} \sim \mathcal{N}(\rho_v^{(0)}, \sigma_\rho^2 I)$, where $\sigma_\rho^2$ is a fixed hyperparameter. Thus, we replace the EFE objective function in Eq. 3 with

$$\mathcal{L}_{\text{hier}} = \log p(\alpha) + \log p(\rho^{(0)}) + \sum_s \log p(\rho^{(s)} \mid \rho^{(0)}) + \sum_{v,i} \log p\left(x_{vi} \mid x_{c_{vi}}; \alpha, \rho\right). \quad (4)$$

where the $\ell_2$-regularization term now applies only on $\alpha_v$ and the global vectors $\rho_v^{(0)}$.

Fitting the hierarchical model involves maximizing Eq. 4 with respect to $\alpha_v$, $\rho_v^{(0)}$, and $\rho_v^{(s)}$. We note that we have not reduced the number of parameters to be inferred; rather, we tie them together through a common prior distribution. We use stochastic gradient ascent to maximize Eq. 4.

**Amortization.** The idea of amortization has been applied in the literature to develop amortized inference algorithms (Dayan et al., 1995; Gershman and Goodman, 2014). The main insight behind amortization is to reuse inferences about past experiences when presented with a new task, leveraging the accumulated knowledge to quickly solve the new problem. Here, we use amortization to control the number of parameters of the S-EFE model. In particular, we set the per-group embeddings $\rho_v^{(s)}$ to be the output of a deterministic function of the global embedding vectors, $\rho_v^{(s)} = f_s(\rho_v^{(0)})$. We use a different function $f_s(\cdot)$ for each group $s$, and we parameterize them using neural networks, similarly to other works on amortized inference (Korattikara et al., 2015; Kingma and Welling, 2014; Rezende et al., 2014; Mnih and Gregor, 2014). Unlike standard uses of amortized inference, in S-EFE the

input to the functions $f_s(\cdot)$ is unobserved and must be estimated together with the parameters of the functions $f_s(\cdot)$.

Depending on the architecture of the neural networks, the amortization can significantly reduce the number of parameters in the model (as compared to the non-amortized model), while still having the flexibility to model different embedding vectors for each group. The number of parameters in the S-EFE model is $KL(S + 1)$, where $S$ is the number of groups, $K$ is the dimensionality of the embedding vectors, and $L$ is the number of objects (e.g., the vocabulary size). With amortization, we reduce the number of parameters to $2KL + SP$, where $P$ is the number of parameters of the neural network. Since typically $L \gg P$, this corresponds to a significant reduction in the number of parameters, even when $P$ scales linearly with $K$.

In the amortized S-EFE model, we need to introduce a new set of parameters $\phi^{(s)} \in \mathbb{R}^P$ for each group $s$, corresponding to the neural network parameters. Given these, the group-specific embedding vectors $\rho_v^{(s)}$ are obtained as

$$\rho_v^{(s)} = f_s(\rho_v^{(0)}) = f(\rho_v^{(0)}; \phi^{(s)}). \tag{5}$$

We compare two architectures for the function $f_s(\cdot)$: fully connected feed-forward neural networks and residual networks (He et al., 2016). For both, we consider one hidden layer with $H$ units. Hence, the network parameters $\phi^{(s)}$ are two weight matrices,

$$\phi^{(s)} = \{W_1^{(s)} \in \mathbb{R}^{H \times K}, W_2^{(s)} \in \mathbb{R}^{K \times H}\}, \tag{6}$$

i.e., $P = 2KH$ parameters. The neural network takes as input the global embedding vector $\rho_v^{(0)}$, and it outputs the group-specific embedding vectors $\rho_v^{(s)}$. The mathematical expression for $\rho_v^{(s)}$ for a feed-forward neural network and a residual network is respectively given by

$$\rho_v^{(s)} = f_{\text{ffnet}}(\rho_v^{(0)}; \phi^{(s)}) = W_2^{(s)} \tanh\left(W_1^{(s)} \rho_v^{(0)}\right), \tag{7}$$

$$\rho_v^{(s)} = f_{\text{resnet}}(\rho_v^{(0)}; \phi^{(s)}) = \rho_v^{(0)} + W_2^{(s)} \tanh\left(W_1^{(s)} \rho_v^{(0)}\right), \tag{8}$$

where we have considered the hyperbolic tangent nonlinearity. The main difference between both network architectures is that the residual network focuses on modeling how the group-specific embedding vectors $\rho_v^{(s)}$ differ from the global vectors $\rho_v^{(0)}$. That is, if all weights were set to 0, the feed-forward network would output 0, while the residual network would output the global vector $\rho_v^{(0)}$ for all groups.

The objective function under amortization is given by

$$\mathcal{L}_{\text{amortiz}} = \log p(\alpha) + \log p(\rho^{(0)}) + \sum_{v,i} \log p\left(x_{vi} \mid x_{c_{vi}}; \alpha, \rho^{(0)}, \phi\right). \tag{9}$$

We maximize this objective with respect to $\alpha_v$, $\rho_v^{(0)}$, and $\phi^{(s)}$ using stochastic gradient ascent. We implement the hierarchical and amortized S-EFE models in TensorFlow (Abadi et al., 2015), which allows us to leverage automatic differentiation.[3]

**Example: structured Bernoulli embeddings for grouped text data.** Here, we consider a set of documents broken down into groups, such as political affiliations or scientific disciplines. We can represent the data as a binary matrix $X$ and a set of group indicators $s_i$. Since only one word can appear in a certain position $i$, the matrix $X$ contains one non-zero element per column. In embedding models, we ignore this one-hot constraint for computational efficiency, and consider that the observations are generated following a set of conditional Bernoulli distributions (Mikolov et al., 2013c; Rudolph et al., 2016). Given that most of the entries in $X$ are zero, embedding models typically downweigh the contribution of the zeros to the objective function. Mikolov et al. (2013c) use negative sampling, which consists in randomly choosing a subset of the zero observations. This corresponds to a biased estimate of the gradient in a Bernoulli exponential family embedding model (Rudolph et al., 2016).

The context $c_{vi}$ is given at each position $i$ by the set of surrounding words in the document, according to a fixed-size window.

| | data | embedding of | groups | grouped by | size |
|---|---|---|---|---|---|
| **ArXiv abstracts** | text | 15k terms | 19 | subject areas | 15M words |
| **Senate speeches** | text | 15k terms | 83 | home state/party | 20M words |
| **Shopping data** | counts | 5.5k items | 12 | months | 0.5M trips |

**Table 1:** Group structure and size of the three corpora analyzed in Section 3.

**Example: structured Poisson embeddings for grouped shopping data.** EFE and S-EFE extend to applications beyond text and we use S-EFE to model supermarket purchases broken down by month. For each market basket $i$, we have access to the month $s_i$ in which that shopping trip happened. Now, the rows of the data matrix $X$ index items, while columns index shopping trips. Each element $x_{vi}$ denotes the number of units of item $v$ purchased at trip $i$. Unlike text, each column of $X$ may contain more than one non-zero element. The context $c_{vi}$ corresponds to the set of items purchased in trip $i$, excluding $v$.

In this case, we use the Poisson conditional distribution, which is more appropriate for count data. In Poisson S-EFE, we also downweigh the contribution of the zeros in the objective function, which provides better results because it allows the inference to focus on the positive signal of the actual purchases (Rudolph et al., 2016; Mikolov et al., 2013c).

## 3 Empirical Study

In this section, we describe the experimental study. We fit the S-EFE model on three datasets and compare it against the EFE (Rudolph et al., 2016). Our quantitative results show that sharing the context vectors provides better results, and that amortization and hierarchical structure give further improvements.

**Data.** We apply the S-EFE on three datasets: ArXiv papers, U.S. Senate speeches, and purchases on supermarket grocery shopping data. We describe these datasets below, and we provide a summary of the datasets in Table 1.

*ArXiv papers:* This dataset contains the abstracts of papers published on the ArXiv under the 19 different tags between April 2007 and June 2015. We treat each tag as a group and fit S-EFE with the goal of uncovering which words have the strongest shift in usage. We split the abstracts into training, validation, and test sets, with proportions of 80%, 10%, and 10%, respectively.

*Senate speeches:* This dataset contains U.S. Senate speeches from 1994 to mid 2009. In contrast to the ArXiv collection, it is a transcript of spoken language. We group the data into state of origin of the speaker and his or her party affiliation. Only affiliations with the Republican and Democratic Party are considered. As a result, there are 83 groups (Republicans from Alabama, Democrats from Alabama, Republicans from Arkansas, etc.). Some of the state/party combinations are not available in the data, as some of the 50 states have only had Senators with the same party affiliation. We split the speeches into training (80%), validation (10%), and testing (10%).

*Grocery shopping data:* This dataset contains the purchases of $3,206$ customers. The data covers a period of 97 weeks. After removing low-frequency items, the data contains $5,590$ unique items at the UPC (Universal Product Code) level. We split the data into a training, test, and validation sets, with proportions of 90%, 5%, and 5%, respectively. The training data contains $515,867$ shopping trips and $5,370,623$ purchases in total.

For the text corpora, we fix the vocabulary to the 15k most frequent terms and remove all words that are not in the vocabulary. Following Mikolov et al. (2013c), we additionally remove each word with probability $1 - \sqrt{10^{-5}/f_v}$, where $f_v$ is the word frequency. This downsamples especially the frequent words and speeds up training. (Sizes reported in Table 1 are the number of words remaining after preprocessing.)

**Models.** Our goal is to fit the S-EFE model on these datasets. For the text data, we use the Bernoulli distribution as the conditional exponential family, while for the shopping data we use the Poisson distribution, which is more appropriate for count data.

On each dataset, we compare four approaches based on S-EFE with two EFE (Rudolph et al., 2016) baselines. All are fit using stochastic gradient descent (SGD) (Robbins and Monro, 1951). In particular, we compare the following methods:

- A global EFE model, which cannot capture group structure.
- Separate EFE models, fitted independently on each group.
- (this paper) S-EFE without hierarchical structure or amortization.
- (this paper) S-EFE with hierarchical group structure.
- (this paper) S-EFE, amortized with a feed-forward neural network (Eq. 7).
- (this paper) S-EFE, amortized using a residual network (Eq. 8).

**Experimental setup and hyperparameters.** For text we set the dimension of the embeddings to $K = 100$, the number of hidden units to $H = 25$, and we experiment with two context sizes, 2 and 8.[4] In the shopping data, we use $K = 50$ and $H = 20$, and we randomly truncate the context of baskets larger than 20 to reduce their size to 20. For both methods, we use 20 negative samples.

For all methods, we subsample minibatches of data in the same manner. Each minibatch contains subsampled observations from all groups and each group is subsampled proportionally to its size. For text, the words subsampled from within a group are consecutive, and for shopping data the observations are sampled at the shopping trip level. This sampling scheme reduces the bias from imbalanced group sizes. For text, we use a minibatch size of $N/10000$, where $N$ is the size of the corpus, and we run 5 passes over the data; for the shopping data we use $N/100$ and run 50 passes. We use the default learning rate setting of TensorFlow for Adam[5] (Kingma and Ba, 2015).

We use the standard initialization schemes for the neural network parameters. The weights are drawn from a uniform distribution bounded at $\pm\sqrt{6}/\sqrt{K+H}$ (Glorot and Bengio, 2010). For the embeddings, we try 3 initialization schemes and choose the best one based on validation error. In particular, these schemes are: (1) all embeddings are drawn from the Gaussian prior implied by the regularizer; (2) the embeddings are initialized from a global embedding; (3) the context vectors are initialized from a global embedding and held constant, while the embeddings vectors are drawn randomly from the prior. Finally, for each method we choose the regularization variance from the set $\{100, 10, 1, 0.1\}$, also based on validation error.

**Runtime.** We implemented all methods in Tensorflow. On the Senate speeches, the runtime of S-EFE is 4.3 times slower than the runtime of global EFE, hierarchical EFE is 4.6 times slower than the runtime of global EFE, and amortized S-EFE is 3.3 times slower than the runtime of global EFE. (The Senate speeches have the most groups and hence the largest difference in runtime between methods.)

**Evaluation metric.** We evaluate the fits by held-out pseudo (log-)likelihood. For each model, we compute the test pseudo log-likelihood, according to the exponential family distribution used (Bernoulli or Poisson). For each test entry, a better model will assign higher probability to the observed word or item, and lower probability to the negative samples. This is a fair metric because the competing methods all produce conditional likelihoods from the same exponential family.[6] To make results comparable, we train and evaluate all methods with the same number of negative samples (20). The reported held out likelihoods give equal weight to the positive and negative samples.

**Quantitative results.** We show the test pseudo log-likelihood of all methods in Table 2 and report that our method outperforms the baseline in all experiments. We find that S-EFE with either hierarchical structure or amortization outperforms the competing methods based on standard EFE highlighted in bold. This is because the global EFE ignores per-group variations, whereas the separate EFE cannot share information across groups. The results of the global EFE baseline are better than fitting separate EFE (the other baseline), but unlike the other methods the global EFE cannot be used for the exploratory analysis of variations across groups. Our results show that using a hierarchical S-EFE is always better than using the simple S-EFE model or fitting a separate EFE on each group. The hierarchical structure helps, especially for the Senate speeches, where the data is divided into many

|  | **ArXiv papers** | **Senate speeches** | **Shopping data** |
|---|---|---|---|
| Global EFE (Rudolph et al., 2016) | $-2.176 \pm 0.005$ | $-2.239 \pm 0.002$ | $-0.772 \pm 0.000$ |
| Separated EFE (Rudolph et al., 2016) | $-2.500 \pm 0.012$ | $-2.915 \pm 0.004$ | $-0.807 \pm 0.002$ |
| S-EFE | $-2.287 \pm 0.007$ | $-2.645 \pm 0.002$ | $-0.770 \pm 0.001$ |
| S-EFE (hierarchical) | $-2.170 \pm 0.003$ | $\mathbf{-2.217 \pm 0.001}$ | $-0.767 \pm 0.000$ |
| S-EFE (amortiz+feedf) | $-2.153 \pm 0.004$ | $-2.484 \pm 0.002$ | $-0.774 \pm 0.000$ |
| S-EFE (amortiz+resnet) | $\mathbf{-2.120 \pm 0.004}$ | $-2.249 \pm 0.002$ | $\mathbf{-0.762 \pm 0.000}$ |

**Table 2:** Test log-likelihood on the three considered datasets. S-EFE consistently achieves the highest held-out likelihood. The competing methods are the global EFE, which can not capture group variations, and the separate EFE, which cannot share information across groups.

groups. Among the amortized S-EFE models we developed, at least amortization with residual networks outperforms the base S-EFE. The advantage of residual networks over feed-forward neural networks is consistent with the results reported by (He et al., 2016).

While both hierarchical S-EFE and amortized S-EFE share information about the embedding of a particular word across groups (through the global embedding $\rho_v^{(0)}$), amortization additionally ties the embeddings of all words within a group (through learning the neural network of that group). We hypothesize that for the Senate speeches, which are split into many groups, this is a strong modeling constraint, while it helps for all other experiments.

**Structured embeddings reveal a spectrum of word usage.** We have motivated S-EFE with the example that the usage of INTELLIGENCE varies by ArXiv category (Figure 1a). We now explain how for each term the per-group embeddings place the groups on a spectrum. For a specific term $v$ we take its embeddings vectors $\{\rho_v^{(s)}\}$ for all groups $s$, and project them onto a one-dimensional space using the first component of principal component analysis (PCA). This is a one-dimensional summary of how close the embeddings of $v$ are across groups. Such comparison is posible because the S-EFE shares the context vectors, which grounds the embedding vectors in a joint space.

The spectrum for the word INTELLIGENCE along its first principal component is the horizontal axis in Figure 1a. The dots are the projections of the group-specific embeddings for that word. (The embeddings come from a fitted S-EFE with feed-forward amortization.) We can see that in an unsupervised manner, the method has placed together groups related to physics on one end on the spectrum, while computer science, statistics and math are on the other end of the spectrum.

To give additional intuition of what the usage of INTELLIGENCE is at different locations on the spectrum, we have listed the $8$ most similar words for the groups computer science (**cs**), quantitative finance (**q-fin**), math (**math**), statistics (**stat**), and nonlinear sciences (**nlin**). Word similarities are computed using cosine distance in the embedding space. Eventhough their embeddings are relatively close to each other on the spectrum, the model has the flexibility to capture high variabilty in the lists of similar words.

**Exploring group variations with structured embeddings.** The result of the S-EFE also allows us to investigate which words have the highest deviation from their average usage for each group. For example, in the Congressional speeches, there are many terms that we do not expect the Senators to use differently (e.g., most stopwords). We might however want to ask a question like "which words do Republicans from Texas use most differently from other Senators?" By suggesting an answer, our method can guide an exploratory data analysis. For each group $s$ (state/party combination), we compute the top 3 words in $\text{argsort}_v \left( ||\rho_v^{(s)} - \frac{1}{S} \sum_{t=1}^{S} \rho_v^{(t)}|| \right)$ from within the top 1k words.

Table 3 shows a summary of our findings (the full table is in the Appendix). According to the S-EFE (with residual network amortization), Republican Senators from Texas use BORDER and the phrase OUR COUNTRY in different contexts than other Senators.

Some of these variations are probably influenced by term frequency, as we expect Democrats from Washington to talk about WASHINGTON more frequently than other states. But we argue that our method provides more insights than a frequency based analysis, as it is also sensitive to the context in which a word appears. For example, WASHINGTON might in some groups be used more often in

| TEXAS | FLORIDA | | IOWA | | WASHINGTON |
|---|---|---|---|---|---|
| border | medicaid | bankruptcy | agriculture | prescription | washington |
| our country | prescription | water | farmers | drug | energy |
| iraq | medicare | waste | food | drugs | oil |

**Table 3:** List of the three most different words for different groups for the Congressional speeches. S-EFE uncovers which words are used most differently by Republican Senators (red) and Democratic Senators (blue) from different states. The complete table is in the Appendix.

the context of PRESIDENT and GEORGE, while in others it might appear in the context of DC and CAPITAL, or it may refer to the state.

## 4 Discussion

We have presented several structured extensions of EFE for modeling grouped data. Hierarchical S-EFE can capture variations in word usage across groups while sharing statistical strength between them through a hierarchical prior. Amortization is an effective way to reduce the number of parameters in the hierarchical model. The amortized S-EFE model leverages the expressive power of neural networks to reduce the number of parameters, while still having the flexibility to capture variations between the embeddings of each group. Below are practical guidelines for choosing a S-EFE.

**How can I fit embeddings that vary across groups of data?** To capture variations across groups, never fit a separate embedding model for each group. We recommend at least sharing the context vectors, as all the S-EFE models do. This ensures that the latent dimensions of the embeddings are aligned across groups. In addition to sharing context vectors, we also recommend sharing statistical strength between the embedding vectors. In this paper we have presented two ways to do so, hierarchical modeling and amortization.

**Should I use a hierarchical prior or amortization?** The answer depends on how many groups the data contain. In our experiments, the hierarchical S-EFE works better when there are many groups. With less groups, the amortized S-EFE works better.

The advantage of the amortized S-EFE is that it has fewer parameters than the hierarchical model, while still having the flexibility to capture across-group variations. The global embeddings in an amortized S-EFE have two roles. They capture the semantic similarities of the words, and they also serve as the input into the amortization networks. Thus, the global embeddings of words with similar pattern of across-group variation need to be in regions of the embedding space that lead to similar modifications by the amortization network. As the number of groups in the data increases, these two roles become harder to balance. We hypothesize that this is why the amortized S-EFE has stronger performance when there are fewer groups.

**Should I use feed-forward or residual networks?** To amortize a S-EFE we recommend residual networks. They perform better than the feed-forward networks in all of our experiments. While the feed-forward network has to output the entire meaning of a word in the group-specific embedding, the residual network only needs the capacity to model how the group-specific embedding differs from the global embedding.

#### Acknowledgements

We thank Elliott Ash and Suresh Naidu for the helpful discussions and for sharing the Senate speeches. This work is supported by NSF IIS-1247664, ONR N00014-11-1-0651, DARPA PPAML FA8750-14-2-0009, DARPA SIMPLEX N66001-15-C-4032, the Alfred P. Sloan Foundation, and the John Simon Guggenheim Foundation. Francisco J. R. Ruiz is supported by the EU H2020 programme (Marie Skłodowska-Curie grant agreement 706760).

## Footnotes

[1]Alternatively, we could share the embedding vectors $\rho_v$ and have group-specific context vectors $\alpha_v^{(s)}$. We did not explore that avenue and leave it for future work.

[2]Another potential advantage of the proposed parameter sharing structure is that, when the context vectors are held fixed, the resulting objective function is convex, by the convexity properties of exponential families (Wainwright and Jordan, 2008).

[3]Code is available at https://github.com/mariru/structured_embeddings

[4]To save space we report results for context size 8 only. Context size 2 shows the same relative performance.

[5]Adam needs to track a history of the gradients for each parameter that is being optimized. One advantage from reducing the number of parameters with amortization is that it results in a reduced computational overhead for Adam (as well as for other adaptive stepsize schedules).

[6]Since we hold out chunks of consecutive words usually both a word and its context are held out. For all methods we have to use the words in the context to compute the conditional likelihoods.

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
