[Reviews · NeurIPS 2017]

Reviewer 1



This work describes a method for learning different word embeddings for different groups in data, such as speeches made by senators in different states or parties. The motivation being that senators from different parties use the same word in different contexts or meanings. The work is generally an incremental extension of EFEs to grouped data where the groups are known a priori. The test set log-likelihoods show that the organisation of the groups do indeed improve the predictive accuracy of the model, however, the baselines are quite weak. The analysis of the learnt variations between embeddings are also interesting though table 3 would benefit from a comparison with the words obtained via frequency analysis between groups as hinted in line 316. The amortization approach is quite interesting and makes this approach practical without a blow up in the number of parameters. Would it be possible to extend this approach to discover the groups in the data? Minor comments: L34: Typo, (that) how L53-54, repeat of lines 43-44. L68: Typo: differntly, wheter L141,143: Bad reference: eq 9 L302: Eventhough

Reviewer 2



This paper presents a word embedding model for grouped data. It extends EFE to learn group-specific embedding vectors, while sharing the same context vector. To handle groups limited data, the authors propose two methods (hierarchical and amortization) to derive group-specific embedding vectors from a shared one. The paper is clearly written, but the novelty is a bit limited since it is an incremental work beyond EFE. 1. From Table 2, there is no clear winner among the three proposed models (hierarchical, amortiz+feedf, and amortiz+resnet), and the performance differences are subtle especially on the shopping data. If one would like to use S-EFE, do you have any practical guidance on choosing the right model? I guess we should prefer amortiz+resnet to amortiz+feedf, since amortiz+resnet always outperforms amortiz+feedf. Line 276 mentions that hierarchical S-EFE works better when there are more groups. Why? 2. Why Separate EFE performs worse than Global EFE? 3. The authors proposed hierarchical and amortization methods for the reasons in lines 128-131. It is interesting to see how S-EFE performs w.r.t. various data sizes. From this experiment, we might understand if S-EFE is going to surpass hierarchical/amortiz S-EFE, provided more and more data. If yes, would it be better to promote S-EFE as the leading method? If not, why? 4. I understand that pseudo log likelihood is a standard metric for evaluating embedding methods. However, it would be great to see how embedding helps in standard language understanding tasks, like text classification, machine translation, etc. 5. lines 141 and 143: Eq. 9 to Eq. 4

Reviewer 3



This paper proposes an extension to exponential family embeddings (EFE) by adding group structures (S-EFE) via hierarchical modeling or amortization. Specifically, S-EFE uses a shared context vector across all groups and a separate embedding vector to construct exponential family embeddings of the dataset. Statistical sharing is further imposed either by using a common prior distribution from which per group embedding vectors (hierarchical model) are drawn from or by learning a (neural network) function to map global embedding vectors to group embedding vectors (amortization). Experiments on a Congressional speech dataset, arXiv dataset, and groceries dataset show that S-EFE outperforms global and separate EFE in terms of test log likelihood. The paper is well written and the motivation is clear. While the extensions (hierarchical model and amortization) are not excitingly novel, I think the proposed method can potentially be useful in various setups, as demonstrated by the authors in their experiments. - For words that are used differently across fields (e.g., intelligence), S-EFE is able to discover variations in its usage. For words that have relatively similar usages across fields, is the method able to keep their embeddings consistent or there are cases when the context vectors are different enough (e.g., due to different topics in different fields) that they confuse the model? - In terms of training time, how much slower is S-EFE compared to EFE? - Experiment results for hierarchical vs. amortized are mixed. Do the authors have any insights on when one is preferred over the other?